# Association of Average Daily Morphine Milligram Equivalents and Falls in Older Adult Chronic Opioid Users

**DOI:** 10.3390/pharmacy12020062

**Published:** 2024-04-03

**Authors:** Stephanie Hwang, Tamera D. Hughes, Joshua Niznik, Stefanie P. Ferreri

**Affiliations:** 1Division of Practice Advancement and Clinical Education, Eshelman School of Pharmacy, The University of North Carolina, Chapel Hill, NC 27599, USA; 2Center for Aging and Health, Division of Geriatric Medicine, School of Medicine, The University of North Carolina, Chapel Hill, NC 27599, USA

**Keywords:** opioids, morphine milligram equivalent, older adults, falls

## Abstract

Opioids remain commonly prescribed in older adults, despite the known association with falls and fall-related injuries. This retrospective cohort study sought to determine the association of opioid use and falls in older adult opioid users. Using a one-year lookback period in electronic health records, daily morphine milligram equivalents (MMEs) were calculated using prescription orders. Fall history was based on patient self-reporting. A receiver operating characteristic (ROC) curve was used to identify the threshold of average daily MMEs at which the likelihood of falls was increased. Older opioid users were most often women and White, with 30% having fallen in the prior year. In ROC analyses (n = 590), the threshold where fall risk increased was 37 MMEs (*p* = 0.07). Older adults prescribed more than 37 MMEs daily may be at increased fall risk and should be targeted for deprescribing interventions. Additionally, analysis on patient characteristics and covariates suggest that sex, age, COPD, sleep apnea, cancer, and psychiatric conditions may indicate an increased risk of falls in older adults taking chronic opioids (*p* < 0.05). Multifactorial interventions may be needed to modify fall risk beyond medication use alone.

## 1. Introduction

Older adults are at an increased risk of medication-related side effects due to the physiologic consequence of aging [1]. Additionally, polypharmacy, where multiple drugs are taken daily, can lead to an increased risk of adverse drug events and geriatric syndromes, including memory loss, constipation, urinary incontinence, and falls [2]. Falls are the leading cause of fatal and nonfatal injuries in older adults [3]. Falls cause more than 2.8 million injuries treated in the emergency department, 800,000 hospitalizations, and 27,000 deaths annually in the United States [4]. In 2018, 27.5% of older adults reported experiencing at least one fall in the year prior, and 10.2% reported a fall-related injury [5].

Effective interventions for falls in clinical settings involve recognizing and addressing modifiable fall risk factors [6]. One such intervention developed by the Centers of Disease Control and Prevention’s Injury Center is the Stopping Elderly Accidents, Deaths, and Injuries (STEADI) toolkit. This toolkit helps providers screen and manage falls in older adults [6,7,8,9]. Drug use is one of the most modifiable factors to reduce falls and fall-related injuries [10]. Many classes of drugs are known to increase the risk of falls [1]. One class of drugs, opioids, is considered high-risk medications. This class of medications is not recommended in older adults, but is still commonly prescribed [1]. The use of opioids in this population suggests a need for greater clinical scrutiny and intervention [11]. Further evidence-based research is needed to make educated decisions to optimize patient care in patients with chronic pain [12].

Guidelines regarding pain contain discourse in care due to different treatment goals: curing the pain, returning to function, alleviating symptoms, and validating patient experience [13]. The 2016 Centers for Disease Control and Prevention (CDC) guidelines for Prescribing Opioids for Chronic Pain recommend using caution when prescribing at least 50 morphine milligram equivalents (MMEs) per day and avoiding dosages at least 90 MMEs per day [14]. MME is a unit that represents the potency of opioid doses relative to morphine and is used when comparing different types and dosages of opioids, as the potency of medications in this medication class may vary. 

When studying the effectiveness of dose on pain control, function, and quality of life, no difference was found between significantly escalating the dose and maintaining the current dose [14]. However, a higher dose increased the risk for opioid use disorder and opioid overdose. In a Veterans Health Administration study, the average daily MMEs in patients who died from an opioid overdose was 98 MMEs, while the average daily MMEs in patients who did not experience a fatal overdose was 48 MMEs [14]. Further evidence-based research is needed to make educated decisions to optimize patient care in patients with chronic pain, particularly as it relates to deprescribing.

Recent literature stratifies opioids based on appropriateness for older adults regarding concurrent medication use, side effects, and opioid tolerance and suggest an increased risk of falls, fall injuries, and fractures with opioid use [15,16]. Yet, opioids are still commonly prescribed in this population—suggesting a need for greater clinical scrutiny and intervention [1]. Lower dosages reduce the risk for overdoses, but characteristics of daily MME dosage and the association with falls in older adults does not exist. Thus, it is unclear whether reducing dosages would significantly improve outcomes for older adults. To date, no studies have examined the dose–response relationship between opioid use and medication-related adverse effects [12]. Thus, the identification of a dose threshold of daily MMEs to decrease the fall risk in older adults warrants further research. Additionally, guidelines do not address how and when to safely deprescribe opioids. The need for clear deprescribing guidelines is crucial in assessing the benefits and risks of taking an opioid. 

The objectives of this study were to (1) describe the population of older adults taking opioids chronically for non-cancer pain and (2) determine the association between the total daily dose of chronic opioids and the occurrence of falls in older adults. Findings from this study will guide future recommendations for MME dosages to prevent falls in older adults. 

## 2. Materials and Methods

### 2.1. Study Design and Setting

This study was conducted as part of a pragmatic randomized trial wherein primary care providers (PCPs) in North Carolina received opioid and benzodiazepine deprescribing recommendations from clinical pharmacists through electronic health record (EHR) communications (NCT04272671) [17]. Data were collected on older adult patients (at least 65 years old) who were ambulatory and seen in one of ten primary care outpatient clinics in the UNC Health Care System. For inclusion in the study, participants took at least one opioid chronically. Patients who underwent active cancer treatment (active chemo, surgery, and radiation), palliative care, and end-of-life treatment were excluded because opioid use may be more appropriate in patients with cancer. 

### 2.2. Data Collection

EHR data from September 2018 to December 2021 were used to identify eligible patients presenting from ten primary care practices enrolled in the study. Patients were eligible for inclusion if they met the criteria for chronic opioid use, defined as having at least four opioid prescriptions in the year prior to the index date with at least one prescription being in the last 90 days. Baseline characteristics were gathered at the time of each patient’s clinic visit (index date) using EHR data. This included age, sex, diagnoses for acute pain, chronic obstructive pulmonary disease (COPD), cardiovascular disease (CVD), sleep apnea, cancer, psychiatric conditions, and concurrent benzodiazepine use, which were used in prior studies on opioid use [12]. 

### 2.3. Primary Exposure

Prescription orders for opioids in the year prior to each patient’s index date were used to calculate exposures. Drug name, strength, directions, and quantity were extracted to calculate average daily milligram exposure and then converted to average daily MMEs using the appropriate conversion factor. The total exposure over the one-year period was calculated as the sum of the average daily MMEs times the days of supply for each order divided by 365 days. 

### 2.4. Primary Outcome

The primary outcome was fall history in the year prior based on the STEADI question “Did you fall in the past year?”, which was integrated into the patient’s EHR. Patients were excluded if their response was not recorded in the patient’s EHR. 

### 2.5. Data Analysis

Histograms and scatterplots (R version 4.0.5 (R Core Team, 2021)) were performed based on patient fall status to explore the relationships between total daily MMEs, fall status, and age. A receiver operating characteristic (ROC) curve was developed to predict the fall status with MMEs and covariates by calculating the area under the curve (AUC). Further statistical analysis to determine the association between fall status, total daily MMEs, sex, age, concurrent benzodiazepine use and co-morbid COPD, CVD, sleep apnea, cancer, and psychiatric indicators included a logistic regression model using the lasso regression analysis method, Pearson’s chi-squared test, and Fisher Exact Test. Covariate adjustments were used to determine the association between daily MMEs and falls in older adults. ROC curve analysis was used to find the MMEs with the largest AUC within the data to identify the threshold. Additional exploratory data analysis was performed on baseline characteristics and covariates using the logistic regression model to identify potential indicators of falls in older adults prescribed chronic opioids. 

### 2.6. IRB and Protection of Human Subjects

The UNC ethics board exempted this study (IRB #20-1904). 

## 3. Results

### 3.1. Participants

A total of 762 patients met inclusion criteria; however, 172 patients (23%) were excluded due to an undocumented fall status in the EHR. Of the 590 patients with a known fall status, 383 patients (65%) did not experience a fall compared to 207 patients (35%) who experienced a fall in the prior year (Table 1). 

### 3.2. Analysis

The average age of older adult patients taking chronic prescription opioids for non-cancer pain was 75 ± 7.4 years. The median total daily MMEs prescribed in adults aged 65 to 69 was 17 MMEs, which was higher than the median total daily MMEs of 10 MMEs in adults aged 90 and older. In patients with a known fall status, there were higher rates of co-morbid conditions in patients who had a fall as compared to patients who did not experience a fall. The rates of acute pain, COPD, CVD, sleep apnea, cancer, and psychiatric conditions were higher in patients who experienced a fall, as compared to patients who did not experience a fall (Table 1). The rate of concurrent benzodiazepine use with opioids was higher in patients who experienced a fall compared to patients who did not fall (21.3% vs. 16.9%). 

No trend in the fall rate was found as MMEs increased. Additionally, no patterns were found between total daily MMEs, fall rate, and patient age. However, the overall fall rate increased with patient age. The average fall rate for adults aged 90 to 94 was higher than that of adults aged 70 to 74 (50% vs. 31%, *p* = 0.006). 

Overall, the average daily prescribed MMEs was a satisfactory predictor of fall status in this sample population (AUC = 0.63). Individuals taking more than 37 MMEs or more had a 47% higher risk of falling compared to patients taking less than 37 MMEs (*p* = 0.07, 95% CI 0.95–2.27). Of the 472 patients prescribed less than 37 MMEs, 315 patients (66.7%) did not fall within the year prior, while 157 patients (33.3%) had a fall. Of the 118 patients prescribed more than 37 MMEs, 68 patients (57.6%) did not fall, while 50 patients (42.4%) had a fall. Further analysis using an ROC curve suggests that 37 MMEs as a threshold is only a weak predictor of falls (AUC = 0.54, Figure 1).

Receiver operating characteristic (ROC) curve for MMEs threshold of 36.5.

A logistic regression model was fitted to predict the fall status based on the total daily MMEs, sex, age, concurrent benzodiazepine use, and co-morbid COPD, CVD, sleep apnea, cancer, and psychiatric indicators. Exploratory analysis on covariates using the logistic regression model suggests that gender, age, COPD, sleep apnea, cancer, and psychiatric conditions may be used as indicators for increased risk of falls in older adults taking chronic opioids (*p* < 0.05, Table 2). 

Based on calculations, concurrent benzodiazepine use did not indicate an increased risk in falls in this sample population (*p* = 0.69). Acute pain and CVD were also weak indicators of falls (*p* > 0.05). 

## 4. Discussion

Older adults in this study were prescribed chronic opioids for noncancer pain, despite CDC and Beer’s criteria recommendations to avoid this class of medications in this age group [1,14]. In a U.S. study between 2008 and 2018, opioid prescription fill rates were disproportionately higher in males and females at least 65 years old and females of all ages between 2008 and 2018 [15,18]. Of the patients with a known fall status in the study, 35% of patients experienced a fall while being prescribed chronic opioids. This is greater than the estimated 25% of older adults that experience a fall each year in the United States [19]. Therefore, interventions are needed to improve the appropriate prescribing and use of opioid medications [20].

Study findings were consistent with the current literature, suggesting an increased risk of falls, fall injuries, and fractures in older adults who use opioids as pain treatment [16]. However, this is the first study to identify a threshold for an increased risk of falls in older adults taking chronic opioids for non-cancer pain. Further research is needed to confirm a clear cutoff, as data suggest that a lower recommended opioid dosage may potentially benefit older adults at risk of experiencing a fall without proportionally affecting patients not at risk for a fall. Previous research identified 90 MMEs as the threshold for a high dosage, but age-group-specific thresholds for other medication-related adverse effects were not determined [14,21]. Comparing single-dose MME thresholds between populations can identify and specify at-risk patient groups. Furthermore, daily-dose MMEs can help target opioid deprescribing among patients at high risk of falls with consideration of other clinical factors, such as initial assessment, indications and monitoring. More information is needed to determine the initial assessment, indications, monitoring, and deprescribing in this age group [22].

Age, gender, and comorbidities may also influence the fall risk, as older patients with certain comorbidities had an increased fall risk. This study indicated that patients who are older and had health problems had an increased risk of falls. The literature suggests that predictors of high-risk opioid prescriptions are particular prior nonopioid prescriptions, medical history, incarceration, and demographics [23]. Comorbidities and mental health conditions in relation to high-risk prescribing practices and sociodemographic factors may indicate future chronic opioid use [24]. Average daily MMEs is not sufficient as a lone predictor of falls in older adults prescribed opioids chronically for non-cancer pain. This study shows that 37 MMEs or more was a weak cutoff predictor. Potentially confounding factors (e.g., use of other fall-risk-increasing drugs, age, and comorbidities) were not statistically controlled for; therefore, future research is warranted to control for these factors to identify a more robust cut-off of MMEs in older adults for an increased risk of falls.

The 2022 CDC opioid prescription guideline update emphasizes an individualized, flexible, and multidisciplinary approach to opioid prescription [25]. NSAIDs and tricyclic, tetracyclic, and SNRI antidepressants may be considered alternate options for pain management. However, guidelines recommend prescribing the lowest effective dose and shortest duration needed. Therapies should be prescribed on a case-by-case basis. NSAIDs should be used with caution in older adults with cardiovascular comorbidities, chronic renal failure, or previous gastrointestinal bleeding and are potentially inappropriate for use in older adults with chronic pain due to a higher risk for adverse events with prolonged use. Additionally, antidepressant medications may increase the risks of confusion and falls [25].

Multifactorial interventions may be needed to modify the fall risk beyond medication use alone. Drug regimen reviews, screening tools, and computer-based strategies with pharmacist-led interventions can help identify and address unnecessary medication use in older adults [26]. The 2022 CDC guidelines suggest implementing interventions to mitigate common risks of opioid therapy in older adults, including risk assessment for falls [25]. Clinicians can also educate older adults receiving opioids to avoid medication-related behaviors that increase risk, such as saving unused medications. Caregivers can play an important role in pain management, particularly in older adults with cognitive impairment.

One limitation identified in this study was an unknown duration of opioid therapy due to data availability. Additionally, participants self-reported falls base on the STEADI question “Did you fall in the past year?”, which was integrated into the patient’s EHR. Further, data from EHR did not contain substantial evidence to suggest opioid tolerance. The results of the study may differ if opioid tolerance based on the treatment duration is treated as a covariate. The results of the study may also differ if opioid tolerance based on the treatment duration is treated as a covariate. A lasso regression was used to minimize the effects of these limitations by adjusting issues of underfitting or overfitting data to reduce variance. Another limitation was that concurrent medications, besides benzodiazepines, that may also contribute to falls were not evaluated. 

Future research should continue to evaluate the effect of opioid use on falls in older adults and the potential influence of modifiable characteristics. Furthermore, analyses that quantify the harms of opioids by identifying dosage thresholds are needed to maximize net benefits for patients. Decisions to discontinue or deprescribe opioids should account for drug efficacy, prescription of high dosages, physical risk factors, concurrent medications, and quality of life [21].

## 5. Conclusions

Older adults taking more than 37 MMEs for non-cancer pain had a 47% greater risk of falling compared to those taking less than 37 MMEs. Prescribing therapies on a case-by-case basis using the lowest effective dose and shortest duration is needed, or alternate options for pain management should be considered. Avoiding opioid doses above 37 MMEs may help reduce the risk of falls in older adults. Additional studies should control for confounding factors (e.g., use of other fall-risk-increasing drugs, age, and comorbidities) to confirm this threshold. Multifactorial interventions are needed to improve the appropriate prescribing and use of opioid medications in older adults.

## Figures and Tables

**Figure 1 pharmacy-12-00062-f001:**
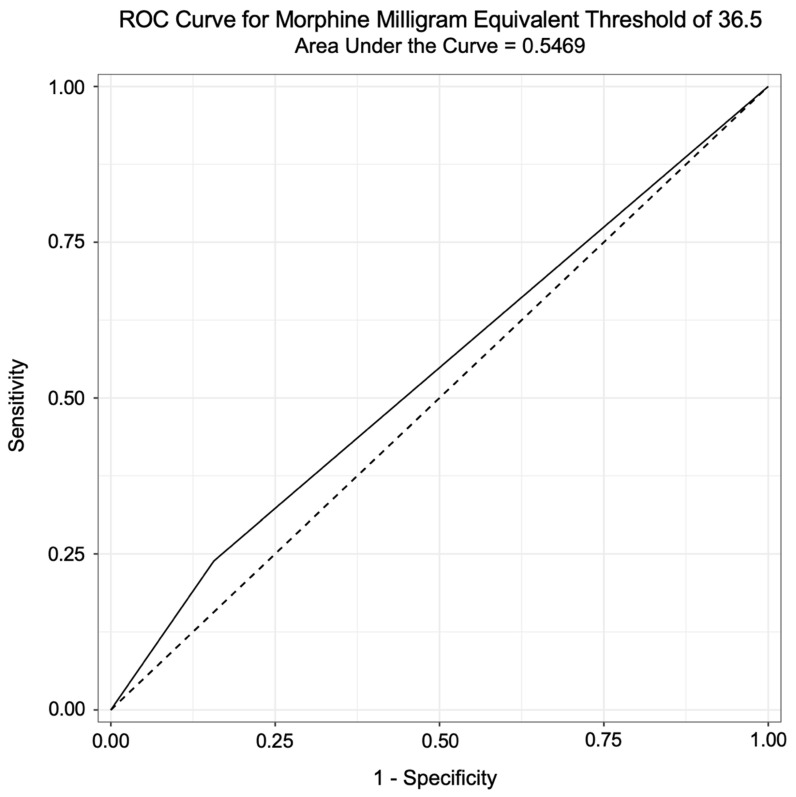
Receiver operating characteristic curve for opioid dosage relative to a patient-reported fall within past year of patient encounter from September 2018 to December 2021.

**Table 1 pharmacy-12-00062-t001:** Baseline characteristics of older adult chronic opioid users using electronic health records (EHR) within the past year of patient encounter from September 2018 to December 2021. Characteristics of Older Adult Patients prescribed opioids for non-cancer pain.

Characteristics	Had a Fall (%) n = 207 (27.1)	Did Not Fall (%) n = 383 (50.3)	Unknown (%) n = 172 (22.6)
DEMOGRAPHIC			
Age (mean (SD))	75.8 (7.6)	74.6 (7.1)	74.6 (7.9)
Male sex	28.5	39.1	29.8
Race			
American Indian or Alaska Native	0.0	1.0	0.0
Asian	0.5	0.5	0.0
Black or African American	12.1	13.5	17.0
Other Race	0.0	0.3	1.2
Patient Refused	0.0	0.3	0.0
Unknown	0.0	0.0	0.6
White	87.4	84.4	81.3
DIAGNOSES			
Acute Pain	98.1	97.7	93.6
COPD	25.6	14.8	19.3
CVD	98.1	94.0	94.2
Sleep Apnea	22.7	16.9	18.7
Cancer	33.3	24.2	21.1
Psychiatric	67.1	52.6	52.6
MEDICATION USE			
MME (mean (SD))	30.1 (67.2)	24.2 (41.9)	31.4 (58.1)

**Table 2 pharmacy-12-00062-t002:** A logistic regression model was fitted to predict the fall status based on covariates. Sex, age, COPD, sleep apnea, cancer, and psychiatric conditions may increase the risk of falls in older adults taking chronic opioids. Concurrent benzodiazepine use did not indicate an increased risk of falls in this sample population (*p* = 0.69). Acute pain and CVD were also weak indicators of falls. Logistic regression model of covariates to predict the fall status.

		95% CIs		
Predictor Variable	Odds Ratio *	Lower	Upper	*p*-Value
DEMOGRAPHIC				
Age	1.10	0.02	0.18	0.016
Sex	1.01	0.00	0.01	0.022
DIAGNOSES				
Acute Pain	0.95	−0.32	0.21	0.694
COPD	1.13	0.03	0.22	0.014
CVD	1.16	−0.04	0.33	0.120
Sleep Apnea	1.12	0.01	0.21	0.029
Cancer	1.11	0.02	0.19	0.014
Psychiatric	1.13	0.05	0.21	0.002
MEDICATION USE				
Benzodiazepine use	1.02	−0.08	0.12	0.693

* Unadjusted odds ratio.

## Data Availability

Data may be shared upon reasonable request.

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
