# Peer review of "Association of Average Daily Morphine Milligram Equivalents and Falls in Older Adult Chronic Opioid Users"

_pharmacy, 2024, doi:10.3390/pharmacy12020062_

Round 1

Reviewer 1 Report

Comments and Suggestions for Authors

This is a well written article, but needs improvments as follows:

The problem is with the identification of the research method. In the abstract, it has been introduced to be retrospective cohort study, but in the body of the paper under the method, it has been recognised as something (unclear) under a pragmatic randomized trial This discrepancy should be removed. For any research type that you identify, you must fill out the appropriate checklist from Equator and should be attached as supplemnetary file.

EQUATOR Network | Enhancing the QUAlity and Transparency Of Health Research (equator-network.org)

Therefore, recognise the method used in this research and bring all related information about it based on the checklist.

Also, conclusion should contain practical considerations for what should be done really to improve the current condition among the older people with a similar context.

Comments on the Quality of English Language

None. 

Reviewer 2 Report

Comments and Suggestions for Authors

This is an interesting correlation between the use of opioids and falls in older adults. It is to be appreciated the recommendation to use opioid paying attention to not overcome a threshold dose without demonize the use of these drugs. Minor points should be addressed:

- dose is reported as morphine milligram equivalents without to specify the different opioids used. It could be interesting analyze and discuss also this point, did the author note a relationship among some specific opioid drug and falls (e.g. fentanyl more than morphine...etc...)

- only benzodiazepine were considered as other drugs. Did the author explore the other medications that these patients receive? In which manner this bias could be standardized?

- suggesting a dose limitation for opioid use, the authors should consider and discuss the possibility to reduce the dose of opioids by using association that allow to limit the quantity. Please see for example doi: 10.3389/fphar.2018.00473. 

Reviewer 3 Report

Comments and Suggestions for Authors

This manuscript uses a retrospective analysis to explore the association between average daily morphine milligram equivalents and fall risk in older adults with chronic who are chronic opioid users. The information presented in the manuscript represents an important question that needs to be further understood. However, the manuscript has several significant concerns that need to be addressed in the manuscript.

The first area relates to the structure of the introduction. It is not clear why the 3rd and 4th paragraphs of the introduction are needed. It appears that the authors are trying to make the link between chronic pain and morphine milligram equivalents per day to set up the study. However, the flow of the introduction with that material is confusing. It is also not clear how the authors are tying the population of interest to the morphine milligram equivalents. Specifically, what is the association between falls, and morphine milligram equivalents in the proposed study population.

Second, what information related to falls could be available in the electronic medical record. For example, is there any information in the electronic health record that would suggest that the individuals have had a visit to the emergency room for a fall or were hospitalized directly due to a fall. The use of this information could potentially fill the gap in terms of the number of individuals who were excluded because their fall status could not be determined. If this is not available, participant self-report should be addressed as a limitation in the manuscript.

Third in Table1, it would be helpful to provide information about the morphine milligram equivalent average dose plus the standard deviation. This is important because the analysis in this manuscript suggests that the ideal cutoff point associated with increased risk of falls is 37 MMES. However, if none of the patients reached that threshold of 37 morphine milligram equivalents, the conclusions drawn from this research may be questionable because there's no evidence, at least in this sample, that such a high dose of MME's is associated with a fall. Specifically, the authors indicate that there is no trend in fall rates was found as MME's increased and there's no pattern between daily MME's and fall rates or patient age. Related to the results, it is unclear how the findings from the logistic regression are specifically related to the research question.

Round 2

Reviewer 3 Report

Comments and Suggestions for Authors

The response to the review is much appreciated. It helped clarify the issues. It is this reviewers opinion that information about the MMEs should still be presented in Table 1 or at a minimum, that the information in the response be included in the manuscript.

Author Response

Thank you for the second review of the manuscript. The details about the MMEs have been added to table one and a discussion of where the 37 or more for MMEs is describes in 3.2 of the results section.